# Exploring Pathogen Presence Prediction in Pastured Poultry Farms through Transformer-Based Models and Attention Mechanism Explainability

**DOI:** 10.3390/microorganisms12071274

**Published:** 2024-06-23

**Authors:** Athish Ram Das, Nisha Pillai, Bindu Nanduri, Michael J. Rothrock, Mahalingam Ramkumar

**Affiliations:** 1Department of Comparative Biomedical Sciences, College of Veterinary Medicine, Mississippi State University, Starkville, MS 39762, USA; ar2903@msstate.edu (A.R.D.); bnanduri@cvm.msstate.edu (B.N.); 2Department of Computer Science and Engineering, Mississippi State University, Starkville, MS 39762, USA; pillai@cse.msstate.edu; 3Egg Safety and Quality Research Unit, USDA-ARS U.S. National Poultry Research Center, Athens, GA 30605, USA; michael.rothrock@usda.gov

**Keywords:** food safety, pathogen, transformer, PageRank

## Abstract

In this study, we explore how transformer models, which are known for their attention mechanisms, can improve pathogen prediction in pastured poultry farming. By combining farm management practices with microbiome data, our model outperforms traditional prediction methods in terms of the F1 score—an evaluation metric for model performance—thus fulfilling an essential need in predictive microbiology. Additionally, the emphasis is on making our model’s predictions explainable. We introduce a novel approach for identifying feature importance using the model’s attention matrix and the PageRank algorithm, offering insights that enhance our comprehension of established techniques such as DeepLIFT. Our results showcase the efficacy of transformer models in pathogen prediction for food safety and mark a noteworthy contribution to the progress of explainable AI within the biomedical sciences. This study sheds light on the impact of effective farm management practices and highlights the importance of technological advancements in ensuring food safety.

## 1. Introduction

The growing concern over *Salmonella*, *Listeria*, and *Campylobacter* in poultry presents a pressing public health issue, as these pathogens lead to significant food-borne illnesses globally [1,2,3]. Addressing the challenge of detecting and managing these bacteria in pastured poultry farms calls for innovative strategies to ensure food safety and protect public health. Our paper introduces a machine-learning-based approach to improving the detection and control of pathogens in poultry production environments. Additionally, we present a new method in the field of explainable AI [4], offering a clear insight into the complex decision-making processes of advanced transformer models.

At the heart of our research is the microbiome—the diverse community of microorganisms inhabiting specific environments such as the gut of poultry. These microbial populations are pivotal in determining the health and disease susceptibility of their hosts. Some microbiota clusters can affect pathogen levels either by maintaining a balance that inhibits pathogenic growth or by fostering conditions conducive to an increase in pathogens [5]. Understanding these complex interactions is essential for effective pathogen risk management.

Recent research in the field of food safety and microbiome has focused on predictive microbiology [6,7] and understanding the decision-making processes of machine learning models. Standard methods for assessing feature importance, such as SHAP [8] and DeepLIFT [9], are commonly used across various datasets. However, these methods often overlook interactions within the microbiota, which can significantly influence the outcomes of these models. Consequently, simple feature importance analyses may not fully explain the underlying biological processes. By employing transformer models, which are renowned for their ability to detect complex data patterns through attention mechanisms [10], we aim to identify microbiota clusters that indicate the presence or absence of pathogens. These models are particularly adept at uncovering data patterns and dependencies [10], making them well suited for analyzing microbiome data relationships that impact pathogen prevalence.

This study not only aims for improved pathogen prediction accuracy but also highlights the critical role of model explainability in biomedical applications. Explainability ensures the trustworthiness, transparency, and reliability of AI predictions for both poultry farmers and biologists. Our method analyzes the model’s attention matrix and applies the PageRank algorithm [11] to clarify feature significance. Furthermore, we extend explainable AI techniques by applying spectral clustering for data cluster analysis and transforming attention matrices into adjacency matrices for graph-based visualizations, improving our interpretive capabilities for AI decision making.

While exploring attention weights in transformer models to demystify their “black-box” nature is not entirely new [12,13,14], we recognize the substantial potential of these weights. Although past studies suggest that attention weights should not be directly equated with explanations [15], the more recent literature suggests that these values can indeed be utilized for explanation [16,17]. Our approach also does not view them as direct interpretations or explanations; instead, we consider attention values as a step towards explaining a model.

By proposing the pathogen prediction method for pastured poultry farming, our paper contributes to the fields of predictive microbiology and food safety. It also paves the way for future research in explainable AI within biomedical sciences, using attention matrices to unveil the reasoning behind model decisions. The broader implications of our findings offer crucial insights into farm management practices that can drive the establishment of an optimal host microbiome that does not support food-borne pathogens and underscores the importance of novel analytical approaches in promoting food safety.

In summary, the primary objective of our study is to develop machine learning models with enhanced explainability for food safety. This research describes highly accurate machine learning models utilizing transformer architectures and introduces a novel model explanation technique that complements existing interpretation methods and shows significant promise for future applications.

## 2. Dataset

In the introductory section, it was highlighted that our research methodology was tested by combining two distinct datasets: the Farm Management Practices dataset and the Microbiome dataset, both of which were sourced from pastured poultry farms. These datasets were acquired from the foundational work of Hwang et al. and Rothrock et al. [18,19].

### 2.1. Farm Management Practices Dataset

This dataset is based on a longitudinal study conducted from March 2014 to November 2017, which covered 42 broiler flocks across 11 pastured poultry farms in the southeastern United States with flock sizes varying between 25 and 1500 birds. The data encompassed both pre-harvest (Feces and Soil) and post-harvest (Ceca, WCR-F, and WCR-P) samples. Documented by Xinran Xu et al. in 2021 [20], the study collected data on poultry farms using movable pens, which were shifted to new pastures daily. The configuration, number, and application of temporary fencing around the pens varied across the farms.

The data collection encompassed 40 key farm practice variables across the lifecycle of a flock, totaling 160 variables, including metadata. The dataset includes variables for the presence of *Salmonella*, *Listeria*, and *Campylobacter*, which are used as the target variables for our study.

### 2.2. Microbiome Dataset

To generate the Microbiome dataset, 16S rRNA gene high-throughput Illumina sequencing was applied to analyze temporal pre-harvest samples, including feces and soil, from the same 41 pastured poultry flocks. This sequencing facilitated the determination of the relative abundance of operational taxonomic units (OTUs). The unique genera identified within the OTUs served as machine learning predictors for assessing the prevalence of food-borne pathogens—*Salmonella*, *Campylobacter*, and *Listeria*—at various growth stages of poultry, categorized as START (2–4 weeks old), MID (5–7 weeks old), and END (8–11 weeks old).

#### Microbiome Analysis

**DNA Extraction:** DNA was extracted from samples according to a semi-automated hybrid DNA extraction protocol [21]. An enzymatic method based on the QIAamp DNA Stool Mini Kit (QIAGEN, Valencia, CA, USA) was combined with a mechanical method with the FastDNA Spin Kit for Feces (MP Biomedicals, Solon, OH, USA). A QIAcube Robotic Workstation was used to purify DNA using the DNA Stool-Human Stool-Pathogen Detection Protocol. Using a Take3 plate and the Synergy H4 multimode plate reader (BioTek, Winooski, VT, USA), the DNA concentration of each sample was determined spectrophotometrically after purification.

**Illumina MiSeq Library Analysis:** A dataset from the Earth Microbiome Project Laboratory at the U.S. Department of Energy, Argonne National Laboratory (Argonne, IL, USA) was used for library construction and sequencing. The hypervariable V4 domain of the bacterial 16S rRNA gene was amplified using the F515 (5′-CACGGTCGKCGGCGCCATT-3′) and R806 (5′-GGACTACHVGGGTWTCT AAT-3′) primer set, with each primer containing Illumina adapter regions (Illumina, Inc., San Diego, CA, USA) and the reverse primer containing the Golay barcodes to facilitate multiplexing [22]. The Illumina MiSeq platform was used to obtain raw reads. The QIIME v1.9.1 pipeline (Quantitative Insights Into Microbial Ecology) generated and processed 3,297,242 raw sequence reads [23]. R1 reads were filtered for quality, and libraries were split according to the Golay barcode sequences (split library fastq.py script, default parameters). With the usearch option [24] and the pick _otus.py script (-*m* usearch, all other parameters were set to the default), sequences were chimera checked against the gold.fa database (http://drive5.com/uchime/gold.fa, accessed on 20 May 2024) and clustered based on their sequence similarity (97%) into operational taxonomic units (OTUs). For each OTU, a representative sequence was selected using the pick_rep_set.py script (utilizing the most abundant method for picking, all other parameters were set to the default) and used for taxonomic assignment with UCLUST and the Greengenes 13_8 database [25] using assign_taxonomy.py (default parameters). Using PyNAST (http://pynast.sourceforge.net, accessed on 10 December 2014) [26], sequences were aligned (align_seqs.py script, default parameters) and filtered (filter_alignment.py). Following this, a phylogenetic tree was generated using make_phylogeny.py (using the default settings and FastTree).

In the context of machine learning, the dataset incorporates input features derived from two distinct sources: the Farm Management Practices dataset and the Microbiome dataset. Specifically, the Farm Management Practices dataset contributes 157 input features, while the Microbiome dataset provides 1823 input features. The model targets three outcome variables, corresponding to the prevalence of three food-borne pathogens: *Salmonella*, *Campylobacter*, and *Listeria*.

## 3. Method

The methodology of this study is structured into three primary stages: data pre-processing, model architecture design and training, and model decoding. While these stages are commonly found in any machine learning project, our approach necessitates further elaboration due to the specific adaptations made to accommodate the transformer model.

### 3.1. Data Pre-Processing

In addressing the farm management data, which comprise both categorical and continuous variables, a critical step in our data pre-processing involved encoding these variables into fixed-dimension vectors. This encoding is essential for facilitating the computation within the attention mechanism of the transformer model. Drawing insights from the work of Xin Huang [27], we observed that traditionally, attention mechanisms in the context of tabular data primarily accommodated categorical features, leaving continuous features to bypass the attention blocks.

To effectively incorporate continuous features into the attention mechanism, we developed a new encoding technique named “scalar-to-vector encoding”. The details of this method are provided below.

Given a scalar value *S*, a maximum value max, a minimum value min, and a target vector dimension *n*, we first define the bin size as follows:$$binsize=\frac{max-min}{n-1}$$

The bin boundaries spanning the complete range of our data are set according to this binsize. The scalar value *S* is then mapped to a corresponding *n*-dimensional vector *V* where all elements are initially set to zero. To embed *S* into *V*, we identify the immediate lower *L* and upper *U* bin boundaries that *S* falls between. The vector *V* is then updated at indices corresponding to *L* and *U* as follows:$$V_{L}=\frac{U-S}{U-L}$$
$$V_{U}=\frac{S-L}{U-L}$$

All other elements in *V* remain at zero, ensuring that *V* represents the scalar value *S* with respect to its position within the specified range.

This method allows continuous variables to be transformed into a fixed-dimensional vector format, making them compatible for processing through the attention mechanism. This extends the capabilities of the transformer model to accommodate scalar continuous values as well. Our design builds upon the foundational principles outlined by Xin Huang [27], expanding its utility to include continuous data alongside categorical inputs.

### 3.2. Model

Utilizing the scalar-to-vector encoding mentioned in the previous subsection, we merged categorical features from the Farm Management Practices dataset and continuous features from both the Farm Management Practices dataset and the Microbiome dataset, preparing them for introduction into the transformer model, as shown in Figure 1. Adopting the transformer encoder architecture outlined in “Attention is All You Need” by Vaswani et al. (2017) [10], we applied it to predict the binary presence of pathogens.

### 3.3. Model Decoding

In the field of model explanation, Shapley Additive Explanations (SHAP) [8] and Deep Learning Important Features (DeepLIFT) [9] are established methods. Our motivation for creating a new model explainability approach stems from two factors. Firstly, the performance of transformers across different data types has been better, a benefit linked to their attention mechanism. This mechanism implies that analyzing the attention matrix can unlock various possibilities in Explainable AI.

Our transformer model features multiple layers of multi-head attention networks in series, with each layer’s output feeding into the subsequent layer’s input. For our purposes in explainable AI and to keep the complexity manageable, only the attention weights from the first block were used for further analysis. Thus, minor modifications in the formula for computing attention [15] allow us to derive the attention matrices for our analysis; these are represented as follows:$$A\left(Q_{l1},K_{l1},V_{l1}\right)=softmax\left(\frac{Q_{l1}K_{l1}^{T}}{\sqrt{d_{k}}}\right)V_{l1}$$

Here, *A* represents the attention value, and *Q*, *K*, and *V* denote the Query, Key, and Value, respectively, with the subscript l1 signifying that these components are exclusively derived from the first layer. This approach is used to construct the attention matrix essential for our model decoding and is utilized for three distinct analyses.

**Feature Importance with PageRank:** We initially applied this approach to calculate feature importance, a common practice in decoding black-box models. PageRank is an algorithm used by Google Search to rank web pages in their search engine results [11]. It measures the importance of website pages by counting the number and quality of links to a page to determine a rough estimate of a website’s importance. Analogously, by using the PageRank algorithm, the significance of a feature can be calculated based on its attention weights.

The PageRank vector, PR, was determined by solving the following eigenvalue problem:$$PR=\alpha APR+\frac{1-\alpha }{N}1$$
where

PR is the vector of PageRank values for all features;α is the damping factor;*A* is the attention matrix, where element $A_{ij}$ denotes the softmax attention weight of feature *i* to feature *j*;*N* is the total number of features;1 is a vector with all elements equal to 1.

**Spectral Clustering to Identify the Signature of Microbiota:** The attention mechanism simplifies the process of identifying feature clusters. These clusters, which are feature groups affecting the model’s output together, are more directly observable in the attention matrix. The importance of identifying such clusters is particularly relevant in our data, as recognizing microbiota clusters is significant in microbiology. Spectral clustering utilizes the spectral features of attention weights to achieve this goal.

Spectral clustering [28] is a technique used in machine learning and data mining to identify clusters in data based on the spectrum (eigenvalues) of the similarity matrix of the data. It works by transforming the data into a lower-dimensional space in which clusters are more apparent and can be easily identified using traditional clustering techniques such as K-means [29] or HAC (Hierarchical Agglomerative Clustering) [30], the latter of which we used in our study.

Our method involved constructing a similarity matrix, *S*, from the data and then deriving the Laplacian matrix, *L*, from *S*. HAC was then performed in the space spanned by the eigenvectors of *L* corresponding to its smallest eigenvalues.

Additionally, these attention matrices can be mapped out in a graph structure to visualize the interactions among the identified OTUs in the microbiota, which is an interesting research question in microbial ecology.

## 4. Experiments and Results

Our investigation focused on two primary questions: whether a prediction model using a transformer architecture, as depicted in Figure 1, outperforms multi-layer perceptron (MLP) models and whether the attention matrix can offer valuable insights into explainable AI. To address the first question, we carried out extensive testing across several models using our dataset, applying a grid search to fine-tune the hyperparameters. The findings from these tests were used in evaluating the Tab-transformer model’s performance relative to that of MLP models. For the second question, we extracted and analyzed the attention matrix from our model, and a detailed discussion of this analysis is presented in the following sections.

### 4.1. Transformer Model Evaluation

For model evaluation, we conducted a series of detailed experiments involving various combinations of epochs, learning rates, test sizes, dropout values, attention layers, and linear layers. These combinations were applied to predict pathogen presence in the following different contexts:Pre-harvest *Salmonella* samples;Pre-harvest *Listeria* samples;Pre-harvest *CampyCapetown* samples;Post-harvest *Salmonella* samples;Post-harvest *Listeria* samples;Post-harvest *CampyCapetown* samples.

We executed the experiments across the following four distinct model architectures to gauge their effectiveness:Multi-layer perceptron (MLP) - Farm Management Practice Variables.Multi-head transformers without scalar-to-vector embedding—Farm Management Practice variables.Multi-head transformers with scalar-to-vector embedding—Farm Management Practice variables.Multi-head transformers with scalar-to-vector embedding—Farm Management Practice variables and Microbiome.

This approach allowed us to compare the efficiency and accuracy of each architecture under similar experimental conditions.

The results of these experiments are shown in Table 1. The table presents the F1 scores for all the models tested on our dataset, demonstrating the gradual enhancement in performance as the models evolve. Our final architecture incorporates a multi-head transformer with scalar-to-vector encoding for continuous variables. The dataset used for this final model combines poultry management variables and microbiome data.

The results of these experiments demonstrate that the performance of the transformer models is better than that of the multi-layer perceptrons. Moreover, the model gives much better predictions of pathogen presence when the farm management data are combined with the Microbiome dataset. With these results, we can infer that the attention mechanism is working in our model, which is essentially the core of transformer models, and we were ready to further investigate the attention matrix.

### 4.2. PageRank Results and Evaluation

As described in the Methods section, we extracted the attention weights from the first multi-head attention block’s self-attention heads for all test scenarios. We then applied the PageRank algorithm to allocate scores to each feature, with the aggregate of these scores equaling 1 amongst all 1823 microbiota features and around 60 farm variable features. The top PageRank-valued features were identified for validation. The top 10 microbiota features identified by the method for post-harvest *Salmonella* are shown in Figure 2 as an example. Similar tables were generated for pre-harvest and post-harvest *Salmonella*, *Listeria*, and *CampyCapetown*.

The outcomes were verified through both qualitative and quantitative means. For the qualitative evaluation, we cross-referenced the top microbiota features identified by our approach with the existing literature to confirm their associations with probiotic or pathogenic characteristics, as discussed in detail in Section 5. For quantitative evaluation, we used DeepLIFT to rank all features in the dataset and then compared the top 100 features identified by both our method and DeepLIFT. This comparison revealed a significant concurrence between the two methods, with approximately 35% of the top features being recognized by both. The Venn diagram in Figure 3 illustrates some of these results. For a comprehensive view of the findings, please refer to the Appendix A.

### 4.3. Spectral Clustering Results

To determine microbiota clusters within the six experimental scenarios (Section 4.1), we utilized attention matrices. The selection of the number of clusters, a critical hyperparameter in our clustering approach, necessitated a methodical decision-making process. We opted for Hierarchical Agglomerative Clustering (HAC) over K-means due to the clarity provided by HAC’s dendrograms in determining the optimal number of clusters. For each scenario, a dendrogram was generated to establish the number of clusters. Subsequently, we applied HAC with average linkage for feature classification into clusters. With this approach, we were able to identify clusters composed of microbes with similar ecological properties. An example of two clusters identified through this method is shown in Table 2. In one of the clusters, the majority of species, including *Actinobacteria*, *Acidobacteria*, *Bacteroidetes*, *Rhodoplanes*, *Bacillus*, *Myxococcales*, and *Candidatus Nitrososphaera*, are non-pathogenic and beneficial microorganisms primarily distributed in soil and water environments. These groups play crucial roles in ecosystem functioning, such as in nutrient cycling and organic matter decomposition. In contrast, the *Lactobacillales* found in the other cluster are typically associated with environments rich in carbohydrates, including dairy products, fermented foods, the gastrointestinal tract of humans and animals, and plant surfaces.

## 5. Discussion

This research aims to make powerful machine learning models more interpretable and transparent when analyzing complex biological data. By combining techniques to explain these models, using effective transformer neural networks, and applying algorithms such as PageRank, we can gain deeper insights into fundamental biological processes. This approach has wide applications to other animal production systems beyond the pastured poultry production system used in this study. It can also be applied to drug discovery, analyzing evolutionary relationships between species, mining agricultural/biomedical text data, understanding gene regulatory networks, and predicting protein structures. Making these advanced models more interpretable will help advance our understanding of biology and lead to new applications that can improve animal and human health.

This section presents a discussion of our findings and related research, focusing on microbiota features that have been identified as significantly associated with the presence of food-borne pathogens. *Salmonella* and *Campylobacter* are among the most prevalent pathogens associated with poultry and are leading causes of bacterial food-borne illness [31]. Taxonomically, *Salmonella* belongs to the phylum *Proteobacteria*, while *Campylobacter* belongs to the phylum *Epsilonbacteraeota* [32]. Listeria, another food-borne pathogen, is a concern not only in poultry but also in a wide range of other foods, such as dairy and ready-to-eat products, and it belongs to the phylum *Firmicutes* [33]. However, the presence and significance of *Listeria* in poultry production, particularly in pastured systems, is more associated with the environment and processing facilities than with the birds themselves.

### 5.1. Role of Firmicutes

Our research finds *Firmicutes* to be one of the major influencing factors for the prevalence of *Salmonella*, *Listeria*, and *Campylobacter* in both pre-harvest (feces, soil) samples and post-harvest (post-processing) samples. While *Firmicutes* is a large phylum of bacteria that includes many non-pathogenic and beneficial organisms, the environmental dynamics of pastured poultry systems can affect the prevalence of various pathogens, including those not classified under *Firmicutes* [34]. Some beneficial *Firmicutes* in the gut flora of pastured poultry may help in competitively excluding pathogenic bacteria by competing for nutrients and attachment sites in the gastrointestinal tract [35]. Studies show that inoculation with *S. Enteritidis* [36] resulted in significant positive correlations with *Firmicutes*, notably affecting the relative abundance of 18 genera.

We found that the family *Bacillaceae*, belonging to the phylum *Firmicutes* [32], influenced the prevalence of *Salmonella*, *Listeria*, and *Campylobacter* in our experiments. The ability of *Bacillus* strains to produce a wide array of antimicrobial peptides (AMPs) and bacteriocins is crucial in their antagonistic effects against enteropathogenic bacteria in the gastrointestinal tract [37]. Specifically, the growth of *Listeria* is inhibited in contaminated environments or products [38]. *Bacillus subtilis PS-216* showed effective inhibition against *Campylobacter jejuni* under microaerobic conditions, demonstrating its potential as a probiotic that could integrate into the chicken intestinal microbiome and combat *campylobacteriosis* [39]. Bacillus strains can produce extracellular polysaccharides, vitamins, and exoenzymes that support the growth of beneficial microbiota, contributing to a healthier gut environment [40,41] and potentially reducing the colonization of pathogens by strengthening the birds’ natural defenses against infections and possibly reducing pathogen shedding.

Our results show that *Rummeliibacillus*, a Gram-positive rod-shaped bacterium, is associated with *Listeria* and *Campylobacter*. Factors such as soil composition and microbial diversity can either facilitate or limit the survival of *Listeria* [42], indicating that microbial competition, potentially including competition from *Rummeliibacillus*, could influence the prevalence of *Listeria*. Lactic acid bacteria (LAB) [43], including *Lactococcus* [44] and *Lactobacillus* species, are known for their role in producing fermented foods and for their ability to inhibit the growth of pathogenic bacteria through the production of lactic acid, bacteriocins, and AMPs. Acidification of the environment due to lactic acid generation by LAB inhibits the growth of [45,46] *Listeria* and *Salmonella*. *Lactobacillus* species have been extensively studied for their probiotic properties in poultry [47], demonstrating benefits such as reduced *Salmonella* contamination (https://today.uconn.edu/2021/06/probiotic-intervention-to-prevent-salmonella-infection-in-poultry/ accessed on 8 April 2024), improved growth performance, immune enhancement, gut microbe sustainability, and contributions to health. *Lactobacillus* cultures or bacteriocins could be used in rinses or coatings for poultry meat post-processing to reduce surface contamination by *Salmonella* [48].

Our results indicate the influence of *Lysinibacillus* species on the presence of *Salmonella* and that of the *Planococcaceae* family on the presence of *Listeria* in the pre-harvest phase, and this suggests a multifaceted approach involving the management of animal waste, the monitoring and treatment of irrigation water, and practices to reduce contamination in food production environments. The role of soil-dwelling or fecal bacteria, such as *Lysinibacillus* [49], in influencing these processes, directly or indirectly, through effects on microbial communities remains a critical area for further research. Another influential genus in our findings, *Solibacillus*, a Gram-positive, rod-shaped, spore-forming bacteria, could potentially compete with *Salmonella* for nutrients in the soil, alter soil microbial community composition, and limit the latter’s ability to proliferate.

*Anoxybacillus* is a genus of thermophilic [50], facultatively anaerobic bacteria within the *Firmicutes* phylum that is known for its ability to thrive at high temperatures and for its presence in diverse environments, including hot springs and dairy products. Detecting *Anoxybacillus* in post-harvest environments indicates that higher temperatures or specific nutrient availability could impact the survival or proliferation of pathogens such as *Campylobacter*.

The class *Clostridia*, part of the *Firmicutes* phylum, includes significant food-borne pathogens that impact poultry and can pose risks to human health [51]. Our results indicate that *Clostridia* plays a critical role in the prevalence of *Salmonella* and *Campylobacter* in soil and fecal samples. Conversely, the presence of *Salmonella* and *Listeria* was associated *Clostridia* species in post-processed meat samples. By contributing to the fermentation process and production of short-chain fatty acids (SCFAs) in the gut [52], The *Ruminococcaceae* family, part of the *Clostridia* class, might help create an intestinal environment that is less favorable for *Salmonella* and *Campylobacter* proliferation. The association between the *Syntrophomonas* genus and *Listeria* contamination in our results suggests a potential role for this group of anaerobic, syntrophic bacteria in modulating the prevalence of *Listeria* in the post-processing environment, possibly due to the syntrophic degradation of butyrate in environmental and industrial processes limiting organic waste that can support pathogens such as *Listeria* [53].

### 5.2. Role of *Proteobacteria*

The phylum *Proteobacteria* represents a vast and diverse group of Gram-negative bacteria that are classified into various classes. *Acinetobacter* can be found in soil and animal (including human) feces, though they are not a predominant component of the gut flora [54], so they could indicate exposure to the environment or the consumption of contaminated food or water [55,56,57]. In soil, these bacteria are key players in breaking down organic substances and nutrient recycling [58].

Our previous studies have underscored the potential role of other microbial species, such as *Acinetobacter*, in the ecology of these pathogens. Specifically, our analysis indicates that *Acinetobacter* played a critical role in pre-harvest (fecal and soil) samples where *Salmonella* was detected. Detection of *Salmonella* and *Acinetobacter* from poultry operations in Washington state was reported [59]. Understanding the nature of this relationship and whether it is synergistic or antagonistic could inform the development of targeted interventions and more effective pathogen control strategies on farms.

### 5.3. Clustering to Uncover the Microbial Community Structure


**ClusterA**


***Rummeliibacillus*** is a part of the *Firmicutes* phylum, and these Gram-positive bacteria are known to form endospores, which allow them to survive in harsh environmental conditions. ***Streptococcus*** is another member of the *Firmicutes* phylum; these are Gram-positive cocci known for their role in both health (as part of the normal microbiota) and disease (causing various infections). *Enterococcus*, *Streptococcus*, and members of the *Enterobacteriaceae* family are primarily associated with the gastrointestinal tract of animals and humans [60], playing roles that range from benign colonization to causing serious infections. ***Salinicoccus*** is a genus of Gram-positive, halophilic bacteria that belong to the family *Staphylococcaceae* [61]. These bacteria are typically found in environments with high salt concentrations [62], such as salt lakes, saline soils, and salted food products, and they have potential applications in biotechnology, including the biodegradation of pollutants in saline conditions and the production of enzymes and other bioactive compounds. ***Enterococcus***, including species such as *Enterococcus faecalis*, is a genus of bacteria that are part of the natural microbiota of the human gastrointestinal tract [63] but can also be found in soil, water, food, and decaying vegetation [64]. In soil, *Enterococcus* species may be introduced through the application of animal manure as fertilizer, contributing to the microbial diversity of agricultural soils.


**ClusterB**


All of the groups in this cluster are ubiquitous in the environment and are found in a wide range of habitats from soil and water to extreme environments such as hot springs (*Crenarchaeota*) and acidic mines (***Acidobacteria***). This distribution underlines their adaptability and the vast diversity of metabolic strategies that they have evolved to exploit different ecological niches. They play crucial roles in their respective ecosystems and are involved in nutrient cycling, decomposing organic matter, and, in some cases, forming symbiotic relationships with plants [65] (e.g., certain ***Alphaproteobacteria***, such as *Rhizobia*) or animals. Their metabolic diversity allows them to perform a variety of biochemical processes critical to Earth’s biogeochemical cycles, such as carbon and nitrogen cycling.

## 6. Conclusions

In this study, we set out to investigate two main questions: firstly, whether incorporating a transformer model into our unique framework would significantly enhance our ability to predict pathogen presence in poultry production environments; secondly, whether we could introduce a novel method for determining feature importance using the attention matrix and PageRank. Although our final model showed modest improvements in predictive performance over previous models applied to our dataset, this improvement was attributed more to the inclusion of microbiome data than to the transformer model itself. However, the primary goal of implementing the transformer model was to facilitate our second objective, which indeed yielded promising results. The method of computing feature importance through the attention matrix and PageRank aligned well with DeepLIFT’s findings and was validated as biologically relevant. Additionally, our study introduced an effective method for identifying microbiota signatures using the same attention matrices, yielding meaningful outcomes. Consequently, we view our contribution as a positive step forward in the field of explainable AI, and we anticipate that it will inspire further research in this direction.

## Figures and Tables

**Figure 1 microorganisms-12-01274-f001:**
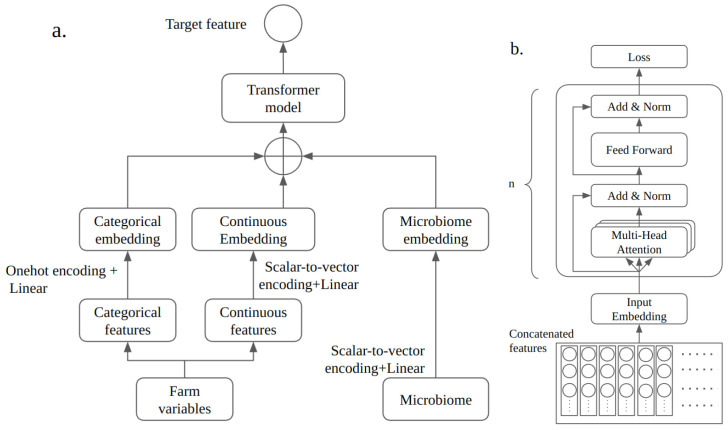
Illustration of the architecture of the transformer model highlighting the integration of both categorical (via one-hot encoding) and continuous (via scalar-to-vector encoding) features into the transformer’s encoder blocks. Panel (**a**) shows the encoding processes, and Panel (**b**) shows the transformer model’s encoder architecture.

**Figure 2 microorganisms-12-01274-f002:**
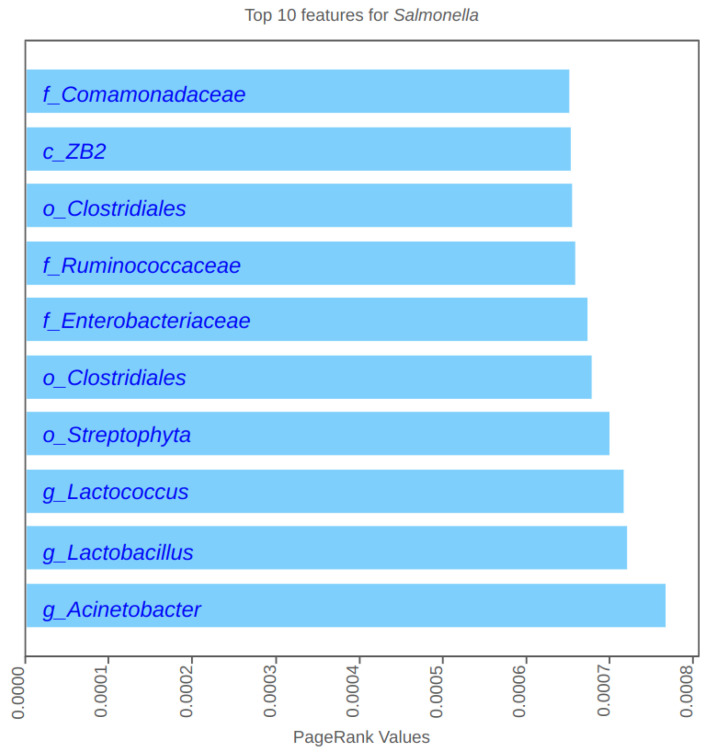
Top 10 most important features recognized by PageRank for post-harvest *Salmonella*.

**Figure 3 microorganisms-12-01274-f003:**
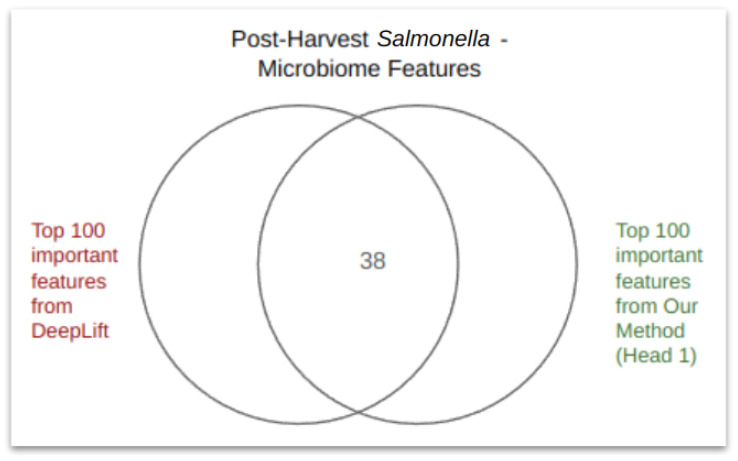
Comparison of the DeepLIFT and PageRank results for post-harvest *Salmonella*. The top 100 features from a total of 1824 were chosen to construct the Venn diagram, which reveals a notable level of agreement between the two methods.

**Table 1 microorganisms-12-01274-t001:** F1 scores for all the models tested on the dataset.

	Pre Harvest			Post Harvest		
	*Salmonella*	*Listeria*	*Campy*	*Salmonella*	*Listeria*	*Campy*
MLP	0.79	0.67	0.84	0.78	0.87	0.95
Multi-Head Transformer w/o scalar-to-vector embedding	0.79	0.72	0.84	0.78	0.92	0.96
Multi-Head Transformer w/ scalar-to-vector embedding	0.78	0.71	0.84	0.83	0.91	0.97
**Multi-Head Transformer w/ scalar-to-vector embedding**	**0.86**	**0.79**	**0.86**	**0.89**	**0.92**	**0.97**

**Table 2 microorganisms-12-01274-t002:** Evaluation of two clusters identified through HAC. Microbes usually found in soil (cluster B/2) and gut (cluster A/1) are accurately identified and grouped separately.

Bacteria	Cluster
*g__Rummeliibacillus*	2
*g__Salinicoccus*	2
*g__Enterococcus*	2
*g__Streptococcus*	2
*f__Enterobacteriaceae;g__*	2
*g__Candidatus Nitrososphaera*	0
*o__iii1-15;f__;g__*	0
*o__RB41;f__;g__*	0
*f__Ellin6075;g__*	0
*f__Gaiellaceae;g__*	0
*o__Solirubrobacterales;f__;g__*	0
*f__Solirubrobacteraceae;g__*	0
*f__Cytophagaceae;g__*	0
*o__Sphingobacteriales;f__;g__*	0
*f__Chitinophagaceae;g__*	0
*g__Bacillus*	0
*g__Rhodoplanes*	0
*o__Myxococcales;f__;g__*	0
*g__DA101*	0

## Data Availability

Restrictions apply to the availability of these data. Data were obtained from the Agricultural Research Service, USDA, and are available from Dr. Rothrock with the permission of the USDA.

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
