# Peer review of "Exploring Pathogen Presence Prediction in Pastured Poultry Farms through Transformer-Based Models and Attention Mechanism Explainability"

_microorganisms, 2024, doi:10.3390/microorganisms12071274_

Round 1
Reviewer 1 Report
Comments and Suggestions for Authors
Introduction. Please described clearly the objectives of the study. The last paragraph of the Introduction is not helpful for readers.
Also in the Introduction, please include a new paragraph to describe in detail the gaps in literature that will be filled by this study. Also, please do make clear the advantages of this work in comparison to other similar studies previously performed by other researchers.
Sections within the manuscript. The division of the manuscript in sections is not in accord with the standard template of the journal. Please clarify with editorial staff of the journal that this approach is acceptable.
Controls. Please describe clearly in a separate sub-section all the controls employed during this study: farms, pathogens, methodologies, reagents.
Visualization. 1) Tables. The inclusion of further tables in the manuscript is encouraged. This will help quick reference of readers to procedures and to findings. 2) Figures. The figures should be enhanced to achieve publication standard.
References. Some recent relevant references are missing, so please do include them and please discuss previous findings in these references in association with current ones.
Conclusions. The concluding section should not include novel ideas; these can be inserted in the Discussion. Also, please tone down the conclusions to bring in line with the findings of the study. Please do not extrapolate.
Author Response
Please find the responses attached with this.

Reviewer 2 Report
Comments and Suggestions for Authors
The manuscript by Athish Ram Das et al. provides interesting results of applying transformer-based models for the prediction of pathogen occurrence in pastured poultry farming. However, the microbiome data used for the machine learning was not processed according to the current research requirements of bioinformatical analysis of microbiota 16S rRNA data.
1. L. 74-75. Please, consider writing what 16S rRNA regions were amplified for the Illumina sequencing.
2. L. 74-87. The authors do not mention any bioinformatical software/pipelines used for the 16S rRNA data processing. How the quality of the sequences was assessed? How the taxonomical identification of the sequences was conducted? What reference database was used?
3. L. 77. Using operational taxonomic units (OTUs) could provide biased results according to recent bioinformatical benchmark studies: https://doi.org/10.1371/journal.pone.0264443, https://doi.org/10.3390%2Fbioengineering9040146, https://doi.org/10.1186/s12864-020-07126-4. Please, consider using amplicon-sequenced variants (ASVs) for the machine learning, as using an ASV-based approach for the differential abundance analysis will provide more robust results.
4. L. 78. How the OTUs prevalence was calculated? Did the authors normalize the data and/or calculate the relative abundances of OTUs for machine learning? What software was used for the prevalence calculation?
5. Figures. Please, write the names of the bacterial families, genera, and species in italics.
6. As the authors used Illumina for high-throughput 16S rRNA sequencing there is an obvious limitation in analyzing bacterial abundance on the species level, however, the authors did not provide any discussion of this limitation in the text of the manuscript. Please, consider writing some discussion on how the high-throughput sequencing of the whole 16S rRNA gene with third-generation technologies would affect the results of your study.
As the whole analysis needs to be revised due to crucial flaws in 16S rRNA data processing, I recommend rejecting the manuscript. The manuscript in its current form does not correspond to the modern requirements for microbiota data analysis.
However, I acknowledge that the authors can redo the whole analysis and resubmit the paper with new results. But this probably will take some time and the resubmitted manuscript will be significantly different from the initial submission.
Author Response

(The authors gave the same response as above.)

Round 2
Reviewer 1 Report
Comments and Suggestions for Authors
The authors have addressed correctly all the points.
No further comments.
Reviewer 2 Report
Comments and Suggestions for Authors
The authors addressed all the comments regarding 16S rRNA sequencing and microbiota data management. Using an OTU-based approach is a major limitation, however, the authors clearly stated that they used previously acquired datasets, which were generated using this approach.
My last comment is the suggestion of using the terms "microbiome" and "microbiota" correctly. When you are showing the results of bacterial presence assessment, the term "microbiota" should be applied, as it represents bacterial communities. The term "microbiome" represents bacterial communities and their "theatre of activity" (https://doi.org/10.1186/s40168-020-00875-0).
